# Alexithymia in Adult Autism Clinic Service-Users: Relationships with Sensory Processing Differences and Mental Health

**DOI:** 10.3390/healthcare11243114

**Published:** 2023-12-07

**Authors:** Emily Josyfon, Debbie Spain, Charlotte Blackmore, Declan Murphy, Bethany Oakley

**Affiliations:** 1Department of Forensic and Neurodevelopmental Sciences, Institute of Psychiatry, Psychology and Neuroscience, King’s College London, De Crespigny Park, London SE5 8AF, UK; 2Social, Genetic and Developmental Psychiatry Centre, Institute of Psychiatry, Psychology and Neuroscience, King’s College London, De Crespigny Park, London SE5 8AF, UK; 3Sackler Institute for Translational Neurodevelopment, King’s College London, De Crespigny Park, London SE5 8AF, UK; 4South London and Maudsley NHS Foundation Trust (SLaM), London SE5 8AZ, UK

**Keywords:** autism, adult, mental health, sensory

## Abstract

Mental health difficulties commonly co-occur with autism, especially in autistic people accessing clinic services, impacting substantially on quality-of-life. Alexithymia (difficulty describing/identifying feelings) and sensory processing differences are prevalent traits in autism that have been associated with depression/anxiety in autistic community samples. However, it is important to better understand interrelationships between these traits in clinical populations to improve identification of service-user needs. In this study, 190 autistic adults (65.3% male), seen in a tertiary autism clinic, completed self-report measures of alexithymia (20-item Toronto Alexithymia Scale), sensory processing differences (Adolescent/Adult Sensory Profile) and depression/anxiety (Hospital Anxiety and Depression Scale). Multiple linear regression models and mediation analyses were used to examine associations between alexithymia, sensory processing differences, and depression/anxiety severity. Across the sample, 66.3% of individuals (*N* = 126) were classified as alexithymic (score ≥ 61). Total alexithymia and difficulty describing/identifying feelings were significantly associated with depression severity (β = 0.30–0.38, highest *p* < 0.002), and difficulty identifying feelings was significantly associated with anxiety severity (β = 0.36, *p* < 0.001). Sensory processing differences were also significantly associated with depression severity (β = 0.29, *p* = 0.002) and anxiety severity across all models (β = 0.34–0.48, highest *p* < 0.001) Finally, difficulty describing/identifying feelings partially mediated the relationships between sensory processing differences and both depression/anxiety severity. Overall, these results highlight that interventions adapted for and targeting emotional awareness and sensory-related uncertainty may improve mental health outcomes in autistic service-users.

## 1. Introduction

### 1.1. Mental Health in Autism

Autism spectrum condition (ASC) (Autism Spectrum Disorder (ASD) is the current DSM-5 classification; however, Autism Spectrum Condition (ASC) and autistic people are terms used interchangeably in the current study, as these two terms encompass the preference of most autistic people and their families in the UK [1,2]) is a neurodevelopmental condition (NDC) characterised by core diagnostic features of social-communication difficulties and restricted and repetitive behaviours—often accompanied by sensory processing differences [3,4], with prevalence of ~1–2% [5,6]. Autistic people (and especially those diagnosed late, in adulthood) report substantially poorer co-occurring mental health outcomes compared to the general population (e.g., depression and anxiety), which negatively impacts their quality of life [7,8,9,10]. Co-occurring mental health difficulties are most notably observed in individuals presenting at specialist ASC clinics seeking diagnosis or treatment, with population:clinical ASC prevalence estimates of 8:14% for depression and 15:26% for anxiety, as compared to 3.3% for depression and 5.9% for anxiety in the general population [11,12]. However, despite the evident need for mental health support, autistic adults describe multiple barriers to accessing mental health interventions, and poorer rates of intervention success, in community and post-diagnostic settings [13,14,15,16,17,18]. One reason posited for these barriers is that autistic and non-autistic communities are often offered the same mental health intervention approaches, despite evidence suggesting subtle mechanistic differences underpinning mental health difficulties experienced between the two groups [19,20]. Therefore, with the objective of informing targeted mental health support, this study aims to interrogate potential mechanisms that may underlie mental health difficulties in individuals presenting at specialist ASC clinics. Specifically, in contrast to most prior research that focuses on community samples of autistic people, here we focus on the interrelationships of alexithymia and sensory processing differences and their associated depression and anxiety severity in an adult clinic population [21,22].

### 1.2. Alexithymia in Autism

Alexithymia is a personality construct comprising three facets: difficulty describing feelings, difficulty identifying feelings and externally orientated thinking (i.e., the tendency to focus away from emotions) [23]. Of note, alexithymia has been found to negatively impact mental health outcomes in the general population [23,24]. Furthermore, alexithymia is estimated to be experienced by 12% of the general population compared to significantly higher estimates of 49.9% of community autistic samples and 55.7–66.2% of autistic adults presenting at specialist ASC clinics [21,25,26,27]. These prevalence estimates of alexithymia in clinical autistic groups are also higher than in other psychiatric groups including in personality, psychotic, mood, and eating disorders: 17–45.7% [28,29,30,31,32].

An aetiological relationship between ASC and alexithymia was first proposed due to similar social difficulties being observed in both autistic and alexithymic individuals [33]. Research with autistic people has suggested that some traits typically associated with ASC—for example differences in social-emotional processing, empathic behaviour, emotion regulation and expression, and difficulties with facial emotion recognition—are influenced by, or entirely explained by, the presence of alexithymia [34,35,36,37,38,39]. Much of this research on alexithymia in ASC has focused on autistic people’s ability to interpret and respond to the emotions of others rather than their own. Despite this, research into internal emotional awareness may be more useful in explaining the high rates of co-occurring mental health difficulties experienced by autistic people, as meta-analytic evidence within the general population has indicated relationships, of medium effect sizes, between emotional awareness and depression/anxiety [40].

### 1.3. Alexithymia and Sensory Processing Differences in Autism

Sensory processing differences are another trait commonly experienced in ASC, that may dually amplify and interact with alexithymia, to further exacerbate mental health difficulties [33,41,42]. Evidence has suggested that 94.4% of autistic adults experience severely altered levels of sensory processing in at least one sensory domain (low registration, sensation seeking/avoiding and sensory sensitivity) [43].

These sensory processing differences may be implicated in alexithymia severity by intensifying uncertainty in processing both external and internal stimuli [22]. Due to intolerance of uncertainty, which is commonly reported by autistic people, this sensory ambiguity, over both internal and external stimuli, may repeatedly be interpreted as threatening and intensify anxiety [44,45,46,47]. Alternatively, sensory processing differences and alexithymia may occur simultaneously and interact to create significant barriers for successful emotion regulation (due to difficulties in both detection and appraisal of internal bodily sensations and their emotional valence, for instance) and thus increasing vulnerability for the development of mental health difficulties [48].

### 1.4. Alexithymia, Sensory Processing Differences and Depression/Anxiety Symptoms in Autism

Integrating the existing evidence described above, recent research has begun to suggest that there may be an interrelationship between alexithymia, sensory processing differences and specific mental health outcomes—namely depression and anxiety—in community ASC samples (i.e., those not currently accessing clinic services; for a summary, see Table 1). For instance, evidence has highlighted that alexithymia may mediate the relationship between autistic traits (specifically, social communication difficulties) and depression/anxiety severity in autistic adults [49,50]. Additionally, alexithymia has been shown to predict anxiety severity longitudinally, over the influence of autistic traits (specifically, social communication difficulties and restricted and repetitive behaviours), in autistic adults [51]. Only one existing study sought to explore sensory processing differences, and reported an association between sensory processing differences, alexithymia, intolerance of uncertainty and anxiety severity in autistic adults [42].

Extending this research to autistic clinic populations is important for identifying appropriate support strategies for individuals actively seeking support, yet only two studies to date have examined associations between alexithymia and depression/anxiety severity in autistic treatment seeking populations (*N* = 122, *N* = 281) [26,27]; and neither of these studies considered the role of sensory processing differences that are hypothesised to interact with alexithymia to worsen mental health outcomes [22,33,41,42,48].

Both studies found that a large proportion of diagnosed adults were classified as severely alexithymic: 66.19% and 55.7% [26,27]. Moreover, they both reported that alexithymia explained the most variance in depression severity, after accounting for autistic traits. In terms of anxiety symptoms, alexithymia was found to be associated with social phobia, but this trend was only apparent without controlling for autistic traits, suggesting that the measure of social phobia (Liebowitz Social Anxiety Scale) utilised was highly overlapping with autism-related social communication difficulties [26]. Neither study investigated the relationship between alexithymia and general anxiety, thus no conclusions can be drawn as to how alexithymia may affect anxiety severity in a clinically presenting autistic group.

Additionally, the studies were limited by measuring alexithymia solely as a unitary construct (i.e., ‘total alexithymia’), despite seminal research suggesting that alexithymia is characterised by several facets (difficulty describing feelings, difficulty identifying feelings, and externally orientated thinking) [23]; and existing evidence that there may be subtle differences in how each of these facets are related to mental health in autistic groups [51].

Thus, it is essential to clarify whether a relationship between alexithymia (also considering its multifaceted nature) and depression/anxiety severity exists in a clinic sample of autistic adults and to consider the influence of sensory processing differences in this context to develop more accurate diagnostic and mental health support options for individuals accessing specialist ASC services.

### 1.5. The Current Study

To our knowledge, this is the first study to date to investigate the severity of alexithymia and consider the mediating effect of alexithymia on a pathway linking sensory processing differences and mental health in service-users presenting at a national adult tertiary autism diagnostic clinic. The current study’s naturalistic sample included males and females between the ages of 18 and 68 years (median = 33 years), included those who met thresholds for referral but did not ultimately receive a diagnosis, and measured alexithymia as a multi-faceted construct, using total and subscale scores from the gold-standard self-report tool: Toronto Alexithymia Scale [52,53]. This naturalistic study design featured only one exclusion criteria—lack of alexithymia data—to ensure the real-world nature of the groups presenting in clinics was captured. To add to the novelty of this study, it was also the first of its kind in the National Health Service (NHS), with previous clinical studies conducted in the context of the German healthcare system.

Study hypotheses were that (1) ASC clinic service-users would experience high rates of alexithymia, (2) alexithymia would be associated with depression/anxiety severity, over the influence of sex, diagnostic status, and sensory processing differences, and (3) alexithymia would be a significant mediator of any relationship found between sensory processing differences and depression/anxiety severity.

## 2. Methods

### 2.1. Procedure

The current study used routinely collected clinical data retrieved from consenting service-users attending the Autism Assessment and Behavioural Genetics Clinic at the South London and Maudsley NHS Foundation Trust. This is a national service that provides assessment for adults with suspected ASC and/or genetic disorders and short-term post-diagnostic support. Autism assessments comprise of a clinical interview, the Autistic Diagnostic Observation Schedule (ADOS), the Autism Diagnostic Interview—Revised (ADI-R) (when possible), multi-disciplinary formulation, and feedback to service-users [54,55]. More detailed clinic methodology is reported elsewhere [56].

All service-users are invited to complete surveys before attending the clinic for the purposes of clinical decision making, and some service-users will also consent for their data to be included in the Developmental Disorders Research Database under a pseudonymised ID code. The collation and use of this database has received ethical approval from London—Southeast Research Ethics committee: 18/LO/0354. The present sample includes all service-users with alexithymia data between February 2014 and October 2019.

### 2.2. Participants

In total, 190 service-users aged between 18 and 68 years were included in this study (see Table 2 and Appendix A for participant characteristics). A total of 153 of these service-users had received an official ASC diagnosis ‘diagnosed’ and 37 were referred for an ASC assessment but were not diagnosed ‘not diagnosed’. We chose to also include those who were not diagnosed in our analyses to maximise both the representativeness of our approaches (i.e., to capture the experiences of individuals who meet thresholds for accessing specialist ASC clinic services but may then ultimately be referred on to other services or discharged), and the potential application of our findings to clinical contexts. Across the sample, 83.2% (*N* = 158) had at least one co-occurring mental health condition. The most common co-occurring group of conditions was anxiety conditions, experienced by 69.5% (*N* = 132) of the sample (see Appendix A).

### 2.3. Measures

#### 2.3.1. Toronto Alexithymia Scale (TAS-20) [57]

The TAS-20 is a self-report alexithymia scale, previously used in studies with autistic people [52]. It features 20 questions and is rated using a 5-point Likert scale from 1 (strongly disagree) to 5 (strongly agree). The items load onto three factors: difficulty describing feelings (5), difficulty identifying feelings (7), and externally orientated thinking (8). Higher scores represent more severe alexithymia and clinical relevance is indicated by scores ≥ 61 [53]. See Appendix A for example items for each measure.

#### 2.3.2. Adolescent/Adult Sensory Profile (AASP) [58]

The AASP is a 60-item self-report scale and was used to capture ASC-related sensory processing differences. Items are rated using a 5-point Likert scale from 1 (almost never) to 5 (almost always). Items load evenly onto four factors: low registration, sensation seeking, sensory sensitivity, and sensation avoiding. Additionally, they cut across six modalities: taste/smell processing, movement processing, visual processing, touch processing, activity level, and auditory processing. In the research literature, a total score has been used as a general indicator of sensory processing differences, by combining the scores across the subscales [59,60]. High and low scores on the subscale are considered atypical—this interpretation was extended to the total score. Specifically, ‘typical’ scores for each factor are low registration (24–35), sensation seeking (43–56), sensory sensitivity (26–41) and sensation avoiding (27–41).

#### 2.3.3. Hospital Anxiety and Depression Scale (HADS) [61]

The HADS is a self-assessment scale that measures depression and anxiety symptom severity in an outpatient setting. It features 16 items which are rated using a 5-point Likert scale from 1 (not at all) to 4 (most of the time). Half the items load onto each factor: depression and anxiety. This scale also does not produce a total score, thus higher scores on each subscale indicate more severe depression/anxiety symptoms. For both scales, scores of 0–7 represent normal, 8–10 represent moderate symptoms, 11–21 represent severe symptoms.

#### 2.3.4. Autism Quotient (AQ) [62]

The AQ is a well-established self-report measure that was used as a broad and dimensional capture of autistic-like traits, rather than a screening or diagnostic tool, as per criticisms of its measurement properties [63,64]. It was included for characterisation purposes but was not used in later modelling due to not measuring our primary interest of sensory processing differences.

#### 2.3.5. Missing Data

Person-mean imputation was used to address missing data and amplify statistical power. The participant’s mean scale score was divided by the completed number of items and the missing items were substituted with this value [65]. A 10% imputation threshold for missing data was used, as evidence has suggested a significant reduction in bias as compared to a higher threshold [66]. Number of participants with imputed items per scale: TAS-20 *N* = 24; AASP *N* = 64; HADS *N* = 11; AQ = 38.

### 2.4. Statistical Analyses

Statistical Package for the Social Sciences (SPSS) Version 29.0.1.0 was used for all analyses [67]. Due to violated normality assumptions, all statistical tests were non-parametric. The severity of alexithymia was characterised using the TAS-20 (total score and subscales) in the whole sample and compared dimensionally (Mann–Whitney tests) and categorically (Chi-squared tests) between the diagnosed and not diagnosed groups.

Mann–Whitney and Spearman’s Rank tests were used to expose (1) any diagnostic group differences in demographic/clinical variables that may account for potential divergence in alexithymia scores and (2) the simple relationships between variables for informing inclusion in regression analyses and later interpretation of findings. The results from these analyses indicated a small number of marginal diagnostic group differences and only one significant difference in sensation seeking behaviour after Bonferroni correction (less sensation seeking behaviours in diagnosed group; *p* < 0.001, *r* = 0.30; see Appendix A). Thus, due to a lack of significant diagnostic group differences, the cohort was included as a single group in later modelling, and marginal group differences were statistically accounted for by including ‘diagnostic status’ as an independent variable in these models. Moreover, non-existent, or weak relationships were found between the main study variables (alexithymia, sensory processing differences and depression/anxiety severity) and age and externally orientated thinking, thus they were excluded as covariates from later regression analyses (see Appendix A). However, sex was found to be significantly associated with sensory processing differences, therefore sex was included as a potential co-variate in later statistical modelling (females experienced more sensory processing differences than males; *p* < 0.001, *r* = 0.29; see Appendix A). Finally, the seven individuals with intellectual disabilities in the cohort were included in all analyses to ensure representativeness of our sample (approximately 30% of autistic people will have a co-occurring intellectual disability according to population estimates; [68]), and as an initial sensitivity analysis indicated no differential effect on the pattern of results by including this group.

Next, a chi-squared test was employed to investigate whether differences in depression and anxiety severity were related to experiencing clinically relevant alexithymia. Additionally, six multiple linear regressions were conducted to investigate interrelationships between alexithymia, sensory processing differences, and depression/anxiety severity over the influence of sex and diagnostic status. Dependent variables for the regressions were either the HADS depression or anxiety subscales. Independent variables were the TAS-20 total score or subscales and AASP total score, controlling for sex and diagnostic status (diagnosed vs. not diagnosed). Secondary analyses additionally controlled for depression/anxiety severity in the models, to determine specificity of associations between alexithymia and anxiety vs. depression symptoms, since though they strongly overlap, they were not multi-collinear (*r* = 0.54). These analyses were powered at 90% and 87%, respectively. The continuous measures were scaled to standardise the variables for the models and the residuals of these models were assessed to ensure that they met statistical assumptions. Bonferroni corrections were applied to all analyses.

Finally, mediation analyses were conducted by using Model 4 in the PROCESS v4.2 macro for SPSS to assess the indirect effects of alexithymia as a mediator on the relationship between sensory processing differences and depression/anxiety severity, controlling for sex and diagnostic status [69]. Alexithymia was inputted as the mediating variable due to theoretical and empirical evidence suggesting that alexithymia may occur in ASC as a result of sensory processing differences distorting the interpretation of environmental inputs [22,42]. The analyses used 95% bias-corrected bootstrap confidence intervals with 5000 resamples. Significant associations were identified by 95% confidence intervals not containing zero.

## 3. Results

### 3.1. Severity of Alexithymia

A total of 66.3% (*N* = 126) of individuals were classified as experiencing clinically relevant alexithymia (score ≥ 61; see Figure 1) [53], with a median of 66 (IQR = 16) on the total scale. Scores were highest for the difficulty identifying feelings subscale (MD = 25, IQR = 9).

### 3.2. Association between Alexithymia and Depression/Anxiety Severity

Across the sample, individuals classified as severely alexithymic (66.3%, *N* = 126) experienced significantly higher levels of depression and anxiety severity as compared to those with lower levels of alexithymia (30.8%, *N* = 56) (depression, *p* = 0.001, *r* = 0.27; anxiety, *p* = 0.005, *r* = 0.23) (see Appendix A).

Dimensional relationships between alexithymia (TAS-20 total score, difficulty describing/identifying feelings subscales) and depression/anxiety (HADS subscales) were also assessed, controlling for potential confounds of sex, diagnostic status, and sensory processing differences (AASP total score).

As demonstrated in Table 3, alexithymia total score (β = 0.32, 95% CI = 0.13–0.50, *p* = 0.001), difficulty describing feelings (β = 0.30, 95% CI = 0.09–0.54, *p* = 0.002) and difficulty identifying feelings (β = 0.38, 95% CI = 0.19–0.57, *p* < 0.001) were significantly associated with depression severity. When, concurrent anxiety severity was controlled for, the associations no longer survived Bonferroni correction but, notably, remained significant at *p* < 0.05 threshold (β = 0.19, 95% CI = 0.02–0.37, *p* = 0.03; β = 0.18, 95% CI = 0.02–0.37, *p* = 0.03; β = 0.20, 95% CI = 0.01–0.38, *p* = 0.04) (see Appendix A).

Moreover, difficulty identifying feelings was significantly associated with anxiety severity (β = 0.36, 95% CI = 0.21–0.58, *p* < 0.001), and similarly, after controlling for concurrent depression severity, the association failed to survive Bonferroni correction, but remained significant at *p* < 0.01 threshold (β = 0.23, 95% CI = 0.06–0.41, *p* = 0.01). For significant associations between alexithymia and mental health outcomes see Figure 2.

### 3.3. Association between Alexithymia, Sensory Processing Differences and Depression/Anxiety Severity

Spearman’s rank correlations revealed that total sensory processing differences were significantly correlated with all facets of alexithymia (TAS-20 total score, difficulty describing/identifying feelings subscales), after correction. The strongest relationship was between difficulty identifying feelings and sensory sensitivity (*r* (125) = 0.51, *p* < 0.001) (see Appendix A).

The regression models demonstrated that sensory processing differences were significantly associated with depression severity over the influence of sex, diagnostic status and difficulty describing feelings (β = 0.29, 95% CI = 0.11–0.48, *p* = 0.002). Despite this, the association did not remain significant after adjusting for concurrent anxiety severity (β = 0.05, 95% CI = −0.13–0.24, *p* = 0.57).

Additionally, sensory processing differences were significantly associated with anxiety severity after accounting for sex, diagnostic status, total alexithymia (β = 0.44, 95% CI = 0.26–0.64, *p* < 0.001), difficulty describing feelings (β = 0.48, 95% CI = 0.31–0.68, *p* < 0.001), and difficulty identifying feelings (β = 0.34, 95% CI = 0.16–0.55, *p* < 0.001) (see Table 3).

These associations with anxiety remained significant after concurrent depression severity was controlled for, over the influence of total alexithymia (β = 0.32, 95% CI = 0.15–0.31, *p* > 0.001) and difficulty describing feelings (β = 0.34, 95% CI = 0.19–0.52, *p* > 0.001), but not difficulty identifying feelings, after Bonferroni correction (β = 0.296 95% CI = 0.10–0.45, *p* = 0.003) (see Appendix A). For significant associations between sensory processing differences and mental health outcomes see Figure 2.

### 3.4. Alexithymia as a Mediator between Sensory Processing Differences and Depression/Anxiety Severity

Following these analyses, relationships between alexithymia, sensory processing differences, and depression/anxiety severity were investigated using mediation analyses, controlling for sex and diagnostic status. The mediation analyses showed that difficulty describing and identifying feelings partially mediated relationships between sensory processing differences and both depression and anxiety severity, respectively (see Figure 3 and Appendix A for more detail).

## 4. Discussion

### 4.1. Severity of Alexithymia

The present study considered the severity of alexithymia and its relationship with sensory processing differences and mental health outcomes in adults accessing a specialist ASC clinic setting. The aim of this approach was to better understand potential mechanisms underlying mental health difficulties to inform support strategies—including for those who did not receive a formal diagnosis of ASC but who may be particularly vulnerable due to limited post-assessment support.

As hypothesised, service-users experienced high rates of alexithymia, as compared to estimates in the wider literature within mixed community-clinic ASC samples and the general population (66.3% vs. 49.9% vs. 12%) [21]. Thus, these findings corroborate that the ability to describe and identify emotions is severely challenging for many ASC clinic service-users and has implications for how to best support these individuals during assessment and post-assessment treatment [26,27]. With regards to assessment, research has indicated that in cases where autism diagnostic observation schedule (ADOS) threshold scores do not align with clinical judgement as to whether an individual should receive a diagnosis, individuals with severe alexithymia were more likely to receive an ASC diagnosis than those with lower levels of alexithymia [70]. Thus, these findings indicate that alexithymia may influence diagnostic decision making, and later care planning, thereby affirming the clear relevance of alexithymia to service access and treatment pathways.

Moreover, with regards the post-assessment treatment, multiple systematic reviews have suggested that identifying and communicating emotions may act as barriers to psychological treatment success for autistic adults, due to a necessity to reflect on and describe emotional states within mental health interventions [17,71]. Despite this, adaptions to cognitive behavioural therapy (CBT)—currently the most widely used psychosocial intervention for autistic adults experiencing for depression and anxiety—have been suggested to lead to increased treatment success compared to standard CBT for depression and anxiety in the limited literature available [72,73]. Important components of these therapies—as noted by clinicians and clinical researchers—are linked to improving emotional awareness by developing (1) a shared vocabulary to describe emotions, (2) skills to identify and anticipate emotional changes and (3) emotion regulation techniques [74]. Future research could consider whether these adaptions may be more widely beneficial across other NDCs in line with evidence highlighting emotional awareness/regulation difficulties as an NDC transdiagnostic feature [75].

### 4.2. Association between Alexithymia and Depression/Anxiety Severity

Aligning with the second study hypothesis, alexithymia total severity, difficulty describing feelings and difficulty identifying feelings was significantly associated with depression severity, and difficulty identifying feelings was significantly associated with anxiety severity—even after adjusting for sex, diagnostic status, and sensory processing differences. However, these associations did not survive controlling for concurrent depression/anxiety severity, possibly due to the high co-occurrence of depression and anxiety in the sample and more broadly. These results replicate and extend previous clinical and community findings, with the novel inclusion of the sensory processing differences variable in analyses [27,51].

Thus, identifying alexithymia as a potential mechanism underlying poor mental health outcomes in individuals with elevated autistic-like traits has clinical implications for ASC clinic service-users and the wider autistic community. Preliminary reviews have indicated that psychological interventions, designed to target alexithymia, but not those intended to treat mental health difficulties, were effective in reducing alexithymia severity in the general population and psychiatric groups—especially mindfulness-based interventions [76,77]. Despite this, the efficacy of these interventions has not yet been explored in autistic adults, but encouraging results exist from mindfulness- and music-based interventions in autistic children [78,79]. Specifically, the Emotional Awareness and Skills Enhancement (EASE) program was designed to improve emotional awareness/acceptance and teach emotion regulation strategies in 16 sessions [78]. The pilot study of the EASE program found that autistic children benefited from improvements in emotional awareness as well as a reduction in depression/anxiety symptoms. With the development of these novel therapeutic approaches, further research is needed to test which approaches might be most appropriate and beneficial for which autistic groups and to target which outcomes. Future investigations, co-produced with autistic communities, can then consider where treatment gaps might exist [80].

### 4.3. Association between Sensory Processing Differences, Alexithymia, and Depression/Anxiety Severity

A novel finding in the current study was that sensory processing differences was significantly associated with depression and anxiety severity over the influence of sex, diagnostic status, and alexithymia in the clinical autistic sample. Further to that, difficulty describing/identifying feelings partially mediated the relationships between sensory processing differences and both depression and anxiety severity. Thus, these findings have implications for unpicking the mechanistic drivers of the interrelationships between sensory processing differences, alexithymia, and mental health outcomes.

The mechanistic processes connecting sensory processing differences, difficulty describing feelings, and depression severity have not been well-characterised in the literature. However, one suggested mechanism that may underpin these relationships is emotion dysregulation—which has been found to inflate the risk for mood conditions in ASC [50,81,82]. Empirically, emotion dysregulation has been highlighted as a mediator between sensory processing differences and behaviour problems, including depressed states, in autistic children and adolescents [83,84]. Similar findings, focusing specifically on negative affect, have also been reported in a sample of non-autistic adults [85]. Additionally, a serial mediation analysis revealed that alexithymia and emotion regulation were strongly associated and mediated the relationship between social responsiveness and depression/anxiety severity [50].

Explanations for the pervasiveness of emotion dysregulation in ASC cite the frequent use of less adaptive cognitive appraisal strategies by autistic people [86]. Namely, autistic adolescents and adults have been found to use more suppression and “others-blame” strategies compared to neurotypical individuals who utilise more reappraisal strategies—and this has been linked to increased depressive symptoms [87,88,89]. Various pilot interventions—including the EASE program (described previously), mindfulness-based interventions and dialectical behaviour therapy—have shown improvements in emotion regulation success in autistic adults, but future research will need to test intervention efficacy on a larger scale [90,91,92]. Additionally, it is important for further research to consider how sensory processing differences and alexithymia may interact to impact emotion regulation and depressive symptoms in ASC, to design more targeted interventions.

The second mediation analysis conducted in the current study identified a relationship between sensory processing differences, difficulty identifying feeling and anxiety severity. A cognitive model of anxiety, that may clarify the mechanisms underpinning this relationship postulates the role of intolerance of uncertainty [22,44,45]. This model states that individuals experiencing an intolerance of uncertainty may repeatedly interpret ambiguous sensory stimuli as threatening, leading to worry and anxiety. To support this theory, empirical evidence has suggested an association between sensory processing differences, alexithymia, intolerance of uncertainty and anxiety severity in autistic adults in the community [42]. Therefore, targeting both alexithymia and intolerance of sensory uncertainty within intervention designs may be beneficial for individuals attending specialist ASC clinics. While incorporating both these mechanisms into an intervention has not yet been investigated, a feasibility trial testing a novel intervention targeting intolerance of uncertainty and related anxiety severity in autistic children was found to be useful and acceptable [93].

However, it is important to note the likely bidirectional relationship between alexithymia and sensory processing differences [33]. As interpreted in this study, alexithymia may be a consequence of the sensory processing differences that are commonly reported in ASC [22,94]. Alternatively, alexithymia may act to disrupt how physiological sensations are processed and subjectively experienced [95]. These varying interpretations suggest that alexithymia may be both a cause and consequence of sensory processing differences, or the processes may act simultaneously to doubly impact mental health outcomes.

We also did not have access in this clinic sample to a robust, standalone measure of restricted and repetitive behaviours, which may further contribute to mechanistic models linking alexithymia with sensory processing differences and mental health. For instance, previous studies in autistic community samples have indicated that restricted and repetitive behaviours are significantly associated with sensory processing differences (including through interrelationships with alexithymia [42], though please also see our previous work showing associations between alexithymia and anxiety symptoms in an autistic community sample after accounting for restricted and repetitive behaviours [51]).

### 4.4. Strengths and Limitations

A major strength of this study was the investigation of alexithymia in the context of the relationship between sensory processing differences and mental health outcomes in a clinical autistic sample. Previous studies in clinical autistic groups focused solely on alexithymia and mental health outcomes, despite evidence for influence of sensory processing differences on anxiety severity [22,41]. The current study adds to the research literature by confirming the severity and role of alexithymia as a partial mediator between sensory processing differences and anxiety in a clinically derived autistic sample. An additional strength of this study is the naturalistic clinical sample which featured only one exclusion criterion: no available TAS-20 data. The aim of this inclusive sample was to inform real-world support strategies for service-users. This approach confirmed that a substantive number of ASC clinic service-users experienced high levels of alexithymia related to sensory processing differences and mental health outcomes; thus, many service-users may benefit from adapted and novel interventions. This finding is especially crucial for referred individuals, not diagnosed with ASC, who are likely to be discharged to community mental health teams that may not be able to sufficiently meet their care needs. Finally, these results are presented as the first clinical study investigating the relationship between alexithymia and mental health in the NHS, adding to previous work within the German healthcare system.

These results must be interpreted in the context of several study limitations. First, while we report statistically significant associations between alexithymia and mental health outcomes, it is important to note that after controlling for concurrent depression/anxiety severity these associations fell to a marginal level of significance. Nevertheless, this result does not undermine the finding that alexithymia may underlie depression and anxiety severity in individuals experiencing elevated autistic-like traits. We also acknowledge that our effect sizes for associations between alexithymia/sensory processing and mental health were relatively low (β = 0.34–0.48), suggesting other relevant factors not measured here contributed to variance in mental health scores (e.g., we know from prior work that genetic, social environmental, cognitive, and physiological/neurobiological mechanisms are all relevant associates of mental health in autism [96]). Due to the cross-sectional nature of the design, we are not able to establish causality between alexithymia and mental health outcomes, but rather the results suggest the relevance of alexithymia as a factor related to both depression and anxiety severity. Future research should develop longitudinal/mechanistic approaches to clarify whether alexithymia modulates anxiety and/or depression severity.

A second limitation derives from the composition of the sample. While the naturalistic sample directly informs the clinical reality of individuals in ASC clinic settings, this approach also encompasses the high amount of individual variation found within clinic samples. Namely, psychiatric comorbidity was common and highly varied within the current sample, leading to a heterogeneous cohort (see Appendix A). Despite this, accounting for the presence of specific mental health diagnoses may not have added value to this research, due the range of experiences already being captured by the dimensional mental health measures used. We also note that, while the decision to include those referred for autism diagnostic assessment but not ultimately diagnosed was important for emphasising that our findings were applicable beyond autistic adults who met threshold for diagnosis (which has implications for informing clinical practice in other service contexts), this group did differ somewhat from the ‘diagnosed’ group. Specifically, there was a nominally higher proportion of females, nominally higher median age, and nominally lower depression and sensory features (except for sensation seeking, which was significantly higher in the not diagnosed vs. diagnosed group). We feel that there is a need for future studies in this area to take a more transdiagnostic approach, with larger samples per diagnostic group, to investigate nuances in associations between alexithymia and mental health and ascertain how support can be better tailored depending on the specific care pathway of the individual.

Finally, we acknowledge the apparent paradox of analysing self-report measures of mental health difficulties in a group of individuals that also report more difficulties describing and identifying their emotions and internal states. One possible explanation as to why autistic adults with high alexithymia are still able to report on their own anxiety and depression symptoms is that there may be a distinction between awareness of one’s difficulties with emotions, compared to awareness of what those emotions are. For instance, agreeing with the statement, “I often don’t know why I am angry” (TAS-20), is somewhat different to agreeing with the statement, “I get sudden feelings of panic” (HADS). The former statement taps the ability to identify emotion triggers, whereas the latter requires endorsement of anxiety, for which the questionnaire itself has applied a specific emotion label. Alternatively, the associations reported here may underestimate the extent of interactions between alexithymia and mental health, as the HADS includes physiological items (“I get a sort of frightened feeling like ‘butterflies’ in the stomach”) and internal emotion sensations may be hardest to report on for autistic adults with higher alexithymia [97]. Addressing measurement issues, such as high reliance on self-report measures, is a key area for future research to test these candidate theoretical explanations [95].

## 5. Conclusions

The prevalence of alexithymia in ASC clinic service-users was high (66.3%), including in those who were referred for assessment but not diagnosed with ASC. Additionally, alexithymia (especially difficulty identifying feelings) and sensory processing differences were significantly associated with depression and anxiety severity, over the influence of sex and autism diagnosis. Finally, difficulty describing/identifying feelings partially mediated the relationship between sensory processing differences and both depression/anxiety severity. Overall, these findings have important clinical implications for adults with elevated autistic traits, indicating that (1) existing psychological therapies adapted for difficulties with emotional awareness may increase accessibility and treatment success for co-occurring mental health conditions, and (2) novel interventions targeting emotional awareness/regulation and sensory-related uncertainty may improve mental health outcomes, and some such approaches are already in development (e.g., [93]). A key recommendation of this work is for further research to identify and directly test adapted/novel interventions for mental health, that integrate understanding of alexithymia and that can be implemented clinically.

## Figures and Tables

**Figure 1 healthcare-11-03114-f001:**
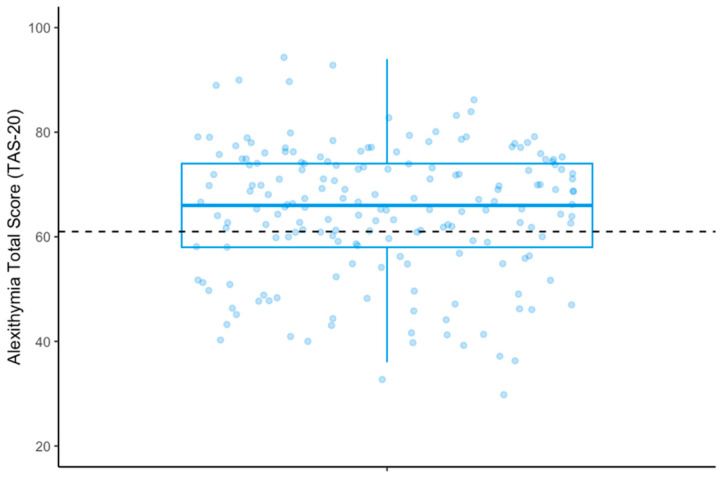
Boxplot showing median and inter-quartile range of total alexithymia scores within the sample. Individual observations are plotted as blue dots to depict the underlying distribution in the sample. The black dashed line represents the threshold score for clinically relevant alexithymia (≥61).

**Figure 2 healthcare-11-03114-f002:**
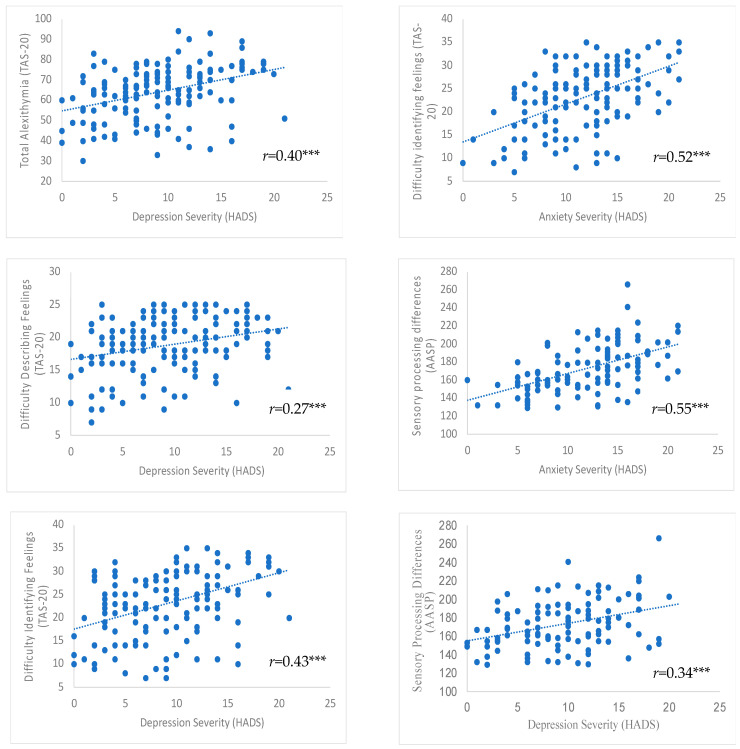
Scatterplots with regression lines fitted, showing the Spearman’s rank correlations between significantly associated variables in the regression models. *r*, Spearman’s rank correlation coefficient; HADS, Hospital Anxiety and Depression Scale; TAS-20, Toronto Alexithymia Scale; AASP, Adolescent/Adult Sensory Profile. *** Bonferroni correction: significant at *p* ≤ 0.001 (0.05/468).

**Figure 3 healthcare-11-03114-f003:**
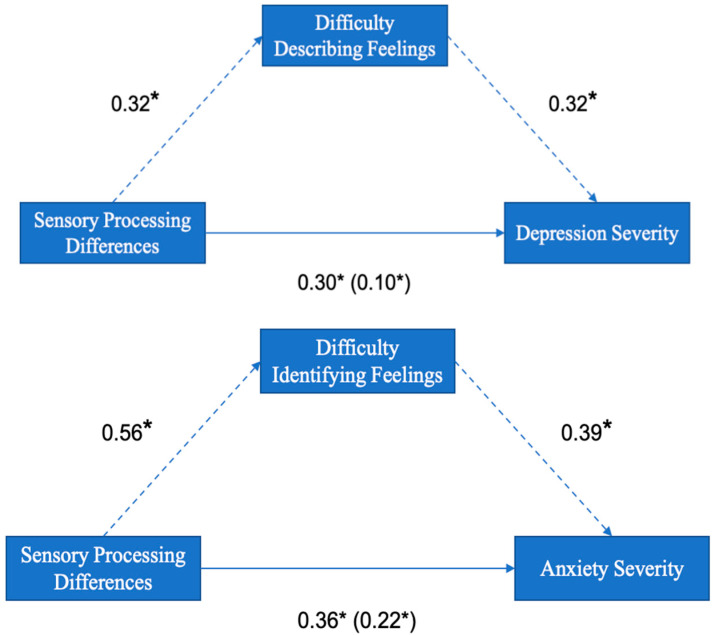
Standardised regression coefficients for the relationships between sensory processing differences and depression/anxiety severity, mediated by difficulty describing/identifying feelings. *n* = 107; The indirect effect of sensory processing differences on depression/anxiety severity via difficulty describing/identifying feelings is shown in parentheses. * Significant at *p* ≤ 0.05.

**Table 1 healthcare-11-03114-t001:** Characteristics and results of studies investigating alexithymia and depression/anxiety severity in autistic people.

Author (Date)	Country	Sample Composition	Sample Size	Age	Sex: Males (Females)	Measure of Alexithymia	Outcome Measures	Percentage Alexithymic (Score ≥ 61)	Relationship between Alexithymia and Mental Health Outcomes
Maisel et al. (2016) [49]	UK and USA	Autistic community	*N* = 76	UK: 26–70 *M* (*SD*) = 44.7 (12.8) USA: 17–36 *M* (*SD*) = 21.7 (4.6)	UK: 30 (10) USA: 29 (7)	TAS-20	Trait anxiety (STAI-T Form-Y) Social anxiety (FNE-Brief) Intolerance of uncertainty (IUS-12) Mindfulness (FFMQ-Nonreactivity) Worrying (PSWQ) Autistic traits (AQ)	Not reported	Alexithymia (*p* < 0.01, β = 0.32) and emotional acceptance (*p* < 0.001, β = −0.45) were moderately associated with anxiety when controlling for autistic traits.
Morie et al. (2019) [50]	UK and USA	Autistic community	*N* = 64	18–65 No descriptive statistics reported	17 (47)	TAS-20	Depression and anxiety (DASS)Emotion regulation (DERS) Autistic traits (SRS-2)	Not reported	Alexithymia and emotion regulation mediated relationships between autistic traits and depression (β = 0.06, se = 0.03, 95% CI 0.02–0.15) and anxiety (β = 0.06, se = 0.02, 95% CI 0.01–0.1).
Oakley et al. (2020) [51]	7 European sites	Autistic community	*N* = 179	12–30 Median (*IQR*) = 19.6 (7.5)	124 (55)	TAS-20	Anxiety (Beck’s anxiety) Depression (Beck’s depression) Autistic traits (SRS-2) (RBS-R)	29.1%	DIF was associated with anxiety symptoms severity cross-sectionally (*p* < 0.001, β = 0.54, 95% CI 0.41–0.77) and longitudinally (*p* = 0.01; β = 0.31, 95% CI 0.08–0.62).
Moore et al. (2021) [42]	UK	Autistic community	*N* = 426	18–77 *M* (*SD*) = 42.8 (13.9)	191 (223)	TAS-20	Depression and anxiety (HADS) Sensory processing differences (SPQ) Autistic traits (SRS-2) (RBQ-2A) Intolerance of uncertainty IUS-12)	61.7%	Mediating effects identified of alexithymia-intolerance of uncertainty-anxiety on relationship between sensory processing and repetitive motor behaviours. (β = 0.0008, se = 0.0005 95% CI 0.0000–0.0021).
Albantakis et al. (2020) [26]	Germany	Autistic service-users	*N* = 122 (confirmed diagnosis)	No age range included *M* (*SD*) = 33.5 (10.4)	83 (39)	TAS-20	Social phobia (LSAS) Depression (BDI-II) Autistic traits (AQ)	55.7%	Alexithymia explained the variance in depression over the influence of autistic traits (β = 0.38, *p* = 0.001). Alexithymia only explained the variance in social phobia without controlling for autistic traits (β = 11.40, *p* = 0.011).
Bloch et al. (2021) [27]	Germany	Autistic service-users	*N* = 281 (confirmed diagnosis)	No age range included *M* (*SD*) = 33.2 (11.0)	219 (62)	TAS-20	Depression (BDI) Autistic traits (AQ)	66.2%	23.2% of variance in BDI explained by autistic traits and alexithymia. DIF was the strongest predictor in the model (GDW = 0.116, 95% CI 0.062–0.173, 50.0% of R2)

*M* (*SD*), Mean (Standard deviation); IQR, Interquartile range; TAS-20, Toronto Alexithymia Scale, STAI-T Form-Y, State Trait Anxiety Inventory Form-Y; FNE-Brief, Fear of Negative Evaluation Scale; IUS-12, Intolerance of Uncertainty Scale; FFMQ-Nonreactivity, Five Facet Mindfulness Questionnaire; PSWQ, Penn State Worry Questionnaire; AQ, Autism Quotient; DASS, Depression, Anxiety and Stress Scale; DERS, Difficulties in Emotion Regulation Scale; SRS-2, Social Responsiveness Scale, Second Edition; RBS-R, Repetitive Behaviour Scale—Revised; SPQ, Sensory Preferences Questionnaire, RBQ-2A, Adult Repetitive Behaviour Questionnaire; BDI, Beck Depression Inventory; LSAS, Liebowitz Social Anxiety Scale; β, regression coefficient; *p*, significance value; se, standard error; 95% CI, 95% confidence interval; GDW; general dominance weights.

**Table 2 healthcare-11-03114-t002:** Whole sample demographic and clinical characteristics.

	Whole Sample		
Demographic/Clinical Measure	*N*	Median (IQR)	Range
Sex: males (females)	124 (66)	-	-
ASC diagnostic status: diagnosed (not diagnosed)	153 (37)		
Presence of intellectual disability	7	-	-
	* **N** *	**Median (IQR)**	**Range**
Age (years)	189	33.0 (20.00)	18–68
Sensory total (AASP)	126	174.0 (37.25)	129–266
Low registration (AASP)	128	43.0 (14.00)	23–71
Sensation seeking (AASP)	125	36.0 (12.25)	19–55
Sensory sensitivity (AASP)	128	46.0 (15.75)	21–75
Sensation avoiding (AASP)	127	49.0 (16.00)	21–74
Depression (HADS)	153	9.0 (7.00)	0–21
Anxiety (HADS)	154	12.0 (6.00)	0–21
Autistic traits (AQ)	152	36.0 (10.00)	11–50
Alexithymia total (TAS-20)	182	66.0 (16.00)	30–94
Difficulty describing feelings (TAS-20)	190	19.0 (6.00)	7–25
Difficulty identifying feelings (TAS-20)	186	25.0 (9.00)	7–35
Externally orientated thinking (TAS-20)	184	23.0 (7.00)	9–36

IQR, interquartile range; ASC, autism spectrum condition; AQ, Autism Quotient; AASP, Adolescent/Adult Sensory Profile; HADS, Hospital Anxiety and Depression Scale; TAS-20, Toronto Alexithymia Scale.

**Table 3 healthcare-11-03114-t003:** Depression and anxiety severity, regressed onto alexithymia severity and sensory processing differences, controlling for sex and diagnostic status.

		Depression (HADS)	Anxiety (HADS)
		β (95% CI)	β (95% CI)
**Model 1 (*N* = 105)**	Sex	−0.12 (−0.64 to 0.13)	−0.01 (−0.38 to 0.38)
	ASC diagnostic status	0.12 (−0.12 to 0.67)	−0.13 (−0.69 to 0.09)
	AASP	0.26 (0.06 to 0.45) **	**0.44 (0.26 to 0.64) *****
	Alexithymia total (TAS-20)	**0.32 (0.13 to 0.50) *****	0.25 (0.07 to 0.45) **
**Model fit (R^2^_adj_)**		F(_4,100_) = 9.26, *p* < 0.001 (24.1%)	F(_4,100_) = 13.23, *p* < 0.001 (32.0%)
**Model 2 (*N* = 107)**	Sex	−0.18 (−0.77 to 0.00) *	−0.05 (−0.50 to 0.25)
	ASC diagnostic status	0.09 (−0.20 to 0.60)	−0.15 (−0.76 to 0.04)
	AASP	**0.29 (0.11 to 0.48) *****	**0.48 (0.31 to 0.68) *****
	Difficulty describing feelings (TAS-20)	**0.30 (0.09 to 0.54) *****	0.23 (0.06 to 0.44) **
**Model fit (R^2^_adj_)**		F(_4,102_) = 9.09, *p* < 0.001 (23.4%)	F(_4,102_) = 12.88, *p* < 0.001 (31.0%)
**Model 3 (*N* = 107)**	Sex	−0.13 (−0.65 to 0.11)	0.00 (−0.36 to 0.36)
	ASC diagnostic status	0.17 (0.01 to 0.80) *	−0.07 (−0.55 to 0.21)
	AASP	0.19 (−0.02 to 0.39)	**0.34 (0.16 to 0.55) *****
	Difficulty identifying feelings (TAS-20)	**0.38 (0.19 to 0.57) *****	**0.36 (0.21 to 0.58) *****
**Model fit (R^2^_adj_)**		F(_4,102_) = 10.55, *p* < 0.001 (26.5%)	F(_4,102_) = 16.86, *p* < 0.001 (37.4%)

β (95% CI), regression coefficient and 95% confidence interval; F(df, df), analysis of variance statistic (regression degrees of freedom, residual degrees of freedom); ASC, Autism spectrum condition; HADS; Hospital Anxiety and Depression Scale; AASP, Adolescent/Adult Sensory Profile; TAS-20, Toronto Alexithymia Scale. * Significant at *p* ≤ 0.05; ** Significant at *p* ≤ 0.01; *** Bonferroni correction: significant at *p* ≤ 0.002 (0.05/24). Bold text is used to emphasise table headers and results that survived Bonferroni correction.

## Data Availability

The participants of this study did not give written consent for their data to be shared publicly, so due to the sensitive nature of the research (clinical data) supporting data is not available.

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
