# Peer review of "Alexithymia in Adult Autism Clinic Service-Users: Relationships with Sensory Processing Differences and Mental Health"

_healthcare, 2023, doi:10.3390/healthcare11243114_

Round 1

Reviewer 1 Report

Comments and Suggestions for Authors

This study examined the association between alexithymia and sensory processing issues, both of which are commonly documented in individuals with autism, with mental health outcomes, specifically depression and anxiety. The intersectionality of variables examined provided a unique perspective in understanding potential barriers to successful outcomes of psychological treatment for mood disorders among adults with autism. The study was very interesting to read and offers critical implications for clinical care, particularly identifying factors that may play a significant role in emotion regulation in this population. The paper was very well-written, clear and organized. The methodology is sound and statistical analyses appropriate and nuanced. The rationale was strong with appropriate literature integrated throughout the introduction to support the current aims. Further, the aims seem to both fill a gap in the research regarding the role of sensory processing in mood disorders in autism, as well as build on prior research by examining anxiety in addition to depression that’s more commonly studied in autism. Another strength of this study is the consideration of a clinical sample of service seekers rather than the general population. In all, I believe the study has a lot to offer both clinicians and researchers dedicated to working with individuals with autism.

The following suggestions are offered to increase clarify in a few places:

1)  It was surprising that the authors chose to include the 37 participants who were referred for assessment but did not receive an autism diagnosis. The authors did later indicate that this small sub-set of participants did not differ significantly from the larger sample in the preliminary analyses, yet the rationale for inclusion is not clear. It also appears in Table S1 that the groups differed on the AASP. What is the benefit for inclusion vs. examining a sample solely of adults with an autism diagnosis?

2)      Similarly, the authors included 7 adults with presence of intellectual disability. Most researchers have this as an exclusionary criteria. This is noted particularly because it is such a small number. If the authors saw an advantage of including those with DD, the rationale as it applies to the study aims was not explained.

3)      The authors conduct a series of analyses with severity for one of the mood symptom (e.g., depression) while controlling for the other (e.g., anxiety), and vice versa. It is well documented that depression and anxiety are highly comorbid. Thus, the fact that controlling for one would lead to insignificant findings of the other seems like a given, as it would be hard to parcel these out. The rationale for this decision is not clear.

Reviewer 2 Report

Comments and Suggestions for Authors

It was my pleasure to review the manuscript “Alexithymia in adult autism clinic service-users: relationships with sensory processing differences and mental health”. The aim of the study was to investigate the severity of alexithymia and consider the mediating effect of alexithymia on a pathway linking sensory processing differences and mental health in service-users presenting at a national adult tertiary autism diagnostic clinic.

 The abstract is well-written and concise.

 The introduction section is well written. In clinical hypothesis formulation variables such as sex, diagnostic status, and sensory processing differences were reported, but not others like repetitive motor behavior, even RMB was mentioned in literature.

 Materials and Methods

Well written

Results are well-presented and instructive.

The discussion well describes the advantages of the study but does not discuss the potential contribution of other factors like repetitive motor behavior on the role of alexithymia on a pathway linking sensory processing differences with depression/anxiety.

 The conclusion is well written. Specifics in describing how novel interventions could target emotional regulation and sensory-related uncertainty would be helpful.

Limitations of the study addressed. 

Reviewer 3 Report

Comments and Suggestions for Authors

Dear authors,

I think your paper is quite interesting, since it deals with the role of Alexithymia in Autistic adults. Carrying out research on this topic is really important in my personal opinion.

The manuscript is clear, pertinent to the field and presented in a well-structured manner.

The manuscript is scientifically robust and the experimental design is appropriate to test the hypothesis. The statistical analyses used are well explained. The study is methodologically appropriate.

The data are interpreted properly and consistently throughout the manuscript.

The conclusions are coherent with the evidence and topics presented.

I would like to suggest you only some minor revision in order to improve this research article.

1.     Despite the term “Autism spectrum condition (ASC)” encompass the preference of most autistic people and their families in the UK, I think that open the article with the most widespread definition of Autism Spectrum Disorder according to DSM-5 is better. I suggest you modify it, or at least integrate.

2.     In the first introduction’s paragraph I suggest exploring what ASD is, in terms of common symptoms, its development throughout life, and the role that early diagnosis during childhood might have on the condition of adult ASD. I provide you some references from which to draw this information:

-       van ’t Hof, M.; Tisseur, C.; van Berckelear-Onnes, I.; van Nieuwenhuyzen, A.; Daniels, A. M.; Deen, M.; Hoek, H. W.; Ester, W. A. Age at Autism Spectrum Disorder Diagnosis: A Systematic Review and Meta-Analysis from 2012 to 2019. Autism 2021, 25 (4), 862–873. https://doi.org/10.1177/1362361320971107.

-       Boccaccio, F. M.; Platania, G. A.; Guerrera, C. S.; Varrasi, S.; Privitera, C. R.; Caponnetto, P.; Pirrone, C.; Castellano, S. Autism Spectrum Disorder: Recommended Psychodiagnostic Tools for Early Diagnosis. Health Psychol Res 2023, 11, 77357. https://doi.org/10.52965/001c.77357.

-       Salari, N.; Rasoulpoor, S.; Rasoulpoor, S.; Shohaimi, S.; Jafarpour, S.; Abdoli, N.; Khaledi-Paveh, B.; Mohammadi, M. The Global Prevalence of Autism Spectrum Disorder: A Comprehensive Systematic Review and Meta-Analysis. Italian Journal of Pediatrics 2022, 48 (1), 112. https://doi.org/10.1186/s13052-022-01310-w.

3.     Although the main values of the regression model are statistically significant, I think it is worth discussing the low score (thus, the strength of the relationship) of the b (beta) value among the limitations.

Great job!
